# Anticardiolipin Antibodies in Patients with Cancer: A Case–Control Study

**DOI:** 10.3390/cancers15072087

**Published:** 2023-03-31

**Authors:** Md. Ashraful Islam Nipu, Shoumik Kundu, Sayeda Sadia Alam, Ashrafun Naher Dina, Md. Ashraful Hasan, Mohammad Khan, Md. Ibrahim Khalil, Tareq Hossan, Md Asiful Islam

**Affiliations:** 1Department of Biochemistry and Molecular Biology, Faculty of Biological Sciences, Jahangirnagar University, Savar, Dhaka 1342, Bangladesh; 2Department of Pharmacology & Therapeutics, Enam Medical College, Savar, Dhaka 1340, Bangladesh; 3New Age Health Science Research Center, Muradpur, Chattogram 4331, Bangladesh; 4Department of Bone Marrow Transplant, Washington University School of Medicine in St. Louis, St. Louis, MO 63110, USA; 5WHO Collaborating Centre for Global Women’s Health, Institute of Metabolism and Systems Research, College of Medical and Dental Sciences, University of Birmingham, Birmingham B15 2TT, UK

**Keywords:** cancer, neoplasms, anticardiolipin antibody, antiphospholipid antibodies, autoimmunity

## Abstract

**Simple Summary:**

Autoimmune diseases cause cancer deaths through thromboembolic events caused by antiphospholipid antibodies. A total of 40 adult cancer patients and 40 age- and sex-matched healthy subjects participated. Blood samples were tested by ELISA methods for anticardiolipin antibodies. Cancer patients had 60.0% (n = 24) aCL antibodies versus none in healthy subjects (*p* < 0.001). All six lung cancer patients had positive aCL antibodies, and colon cancer patients had a borderline significant association (*p* = 0.051). In total, 72.7% of advanced-stage cancer and 81.8% of surgery patients had positive aCL antibodies. Cancer patients with cardiovascular comorbidity had higher aCL antibody positivity (*p* = 0.005).

**Abstract:**

Antiphospholipid antibodies are highly prevalent in autoimmune diseases and mainly associated with thromboembolic events, which is one of the major reasons for cancer-related mortality. Confirmed adult cancer patients were included (n = 40) with an equal number of age- and sex-matched healthy controls. The presence and concentration of anticardiolipin antibodies were investigated by the enzyme-linked immunosorbent assay using the venous blood samples. aCL antibodies were detected in 60.0% (n = 24) of the cancer patients compared to none in the healthy controls (*p* < 0.001). The serum concentration of aCL antibodies was significantly higher in cancer patients than controls (*p* < 0.001) and ranged from 89.0 U/mL to 133.0 U/mL among the aCL-positive patients. All the lung cancer patients (n = 6) were diagnosed with positive aCL, and a borderline significant association of aCL antibody positivity was observed in colon cancer patients (*p* = 0.051). About 72.7% of the advanced-stage cancer individuals and 81.8% of the cancer patients who underwent surgery were diagnosed with positive aCL antibodies. A significant association of aCL antibody positivity was observed with cancer patients comorbid with heart diseases (*p* = 0.005). The prevalence and serum levels of aCL antibodies were significantly higher in cancer patients compared to healthy controls. Cancer patients (i.e., lung, liver, and colon), at advanced-stage, comorbid with heart diseases, who underwent surgery, were more likely to be diagnosed with aCL antibodies.

## 1. Introduction

Cancer is a complex and fatal disease where cells gain the biological capability that enables tumor growth and metastasis through genetic and epigenetic alterations, promoting the uncontrolled proliferation of cells, resistance to cell death, and angiogenesis, resulting in an overall loss of genomic integrity [1]. Despite numerous discoveries of efficacious treatment strategies, cancer is still one of the leading causes of death worldwide. The global estimated incidence of cancer is 12.7 million, which is predicted to rise to 21.4 million by 2030 [1]. In Bangladesh, approximately 200,000 people are newly diagnosed with cancer each year [2,3]. Cancer patients are more prone to endothelial injury, hypercoagulability, and compressed veins, which pathophysiologically stimulate thrombotic complications. Certain anticancer treatment strategies, such as cancer-associated surgery, cytotoxic chemotherapy, and hormone therapy, also facilitate the mechanisms underlying hypercoagulation, leading to thrombotic events as an adverse effect of drugs over the course of treatment. Thrombotic complications, including venous and arterial thrombosis, are among the major causes of cancer-related death globally [4,5]. Genomic profiles of cancer patients identified that tumor-induced mutations in KRAS, STK11, KEAP1, CDKN2B, CTNNB1, and MET are significantly associated with cancer-associated thrombosis [6]. Treating cancer patients with immune checkpoint inhibitors (ICIs) may also result in thrombotic complications in cancer patients, which mainly occur as drug-induced adverse effects. An increase in the incidence of thrombotic events was noticed with prolonged treatment with ICIs [7].

One of the possible mechanisms of exerting thrombotic complications is the procoagulant activities of the tumor cells and interaction with the fibrinolytic system, consequently activating the blood clotting mechanism. As a result, the cancer patients enter a hypercoagulable state [8]. Autoantibodies, including antiphospholipid antibodies (aPLs), are believed to play a role in developing thrombotic events in cancer patients [9,10].

Antiphospholipid antibodies (aPLs) are a group of autoantibodies made up primarily of lupus anticoagulant (LA), anti-2-glycoprotein I (2-GPI), and anticardiolipin [11] antibodies. aPLs react against phospholipids, phospholipid–protein complexes, and phospholipid-binding proteins [11,12]. These antibodies are the characteristic features of antiphospholipid syndrome associated with thrombosis and/or pregnancy complications [12,13]. Some other non-classical aPLs, such as antiphosphatidylinositol, anti-annexin A5 antiphosphatidylserine, antiphosphatidylethanolamine, and antiprothrombin antibodies, also play a role in the pathogenesis of other diseases [14,15]. The presence of the high titers of aPLs is found in certain medical conditions, including migraine, epilepsy, dementia, Behçet’s disease, systemic lupus erythematosus (SLE), and other autoimmune diseases [16,17,18,19].

Although it is presumed that aPLs are associated with the development of thromboembolic complications in solid and hematologic cancers, their relationship is yet to be elucidated [20,21,22]. Among the aPLs, the presence of aCL antibodies has been observed to be significantly higher in cancer patients [22,23,24]. Moreover, the prevalence of thrombotic complications was also observed to be significantly higher in aCL-positive cancer patients than in aCL-negative cancer patients [21,25]. Therefore, the presence of aCL may act as a risk factor in patients with cancer, which may additionally increase the risk of mortality. This study aimed to explore the presence of aCL antibodies in patients with different types of cancer compared to that in healthy control subjects.

## 2. Materials and Methods

### 2.1. Study Population

Confirmed adult cancer patients (≥18 years) were enrolled from the Enam Medical College Hospital, Savar, Dhaka, between 25 January 2020 and 15 December 2020. A control group of age- and sex-matched healthy individuals without cancer and previous history of pregnancy loss, thrombophilia, thrombosis, or autoimmune diseases were selected. 

### 2.2. Data Collection

The clinical assessment was recorded on a standardized data collection sheet that included demographic information, cancer type, stage, ongoing treatments or medications, and comorbidities.

### 2.3. Blood Sampling

Through venipuncture procedure, venous blood (5.0 mL) was collected into a plain tube and centrifuged at 4000 rpm (ROTOFIX 32 A, Andreas Hettich GmbH & Co. KG, Tuttlingen, Germany) for 20 min at 4 °C. Then, 2.0 mL serum sample was taken into an Eppendorf tube with their respective sample ID and was stored at −80 °C until use.

### 2.4. Anticardiolipin Antibody Assay

The quantitative aCL assay (IgG/IgM) was performed following the principle of enzyme-linked immunosorbent assay (ELISA) using a commercially available kit (AESKULISA Cardiolipin check, AESKU Diagnostics, Wendelsheim, Germany). An amount of 100 μL of sample buffer was taken into the Eppendorf tube, and then a 10 μL serum sample was added. The serum sample and the buffer were mixed thoroughly, and the mixture was poured into each well of the microtiter plate, except the last three consisting of the positive, negative, and cut-off control. The microtiter plate was incubated at room temperature (25 °C) for one hour. Following incubation, each well was washed three times with 100 μL wash solution. Then, 100 μL conjugate solution was added into each well, and the plate was again incubated for 30 min at room temperature (25 °C). Each well was washed with 100 μL wash solution (three times). Next, 100 μL substrate solution was added into each well, and the plate was shaken gently, and the final incubation was performed for 15 min. Finally, 100 μL stop solution was added to each of the wells, allowed to mix properly by shaking, and then the optical density was measured at 450 nm versus 620 nm. The results are expressed in U/mL for IgA/G/M, and the cut-off for positive aCL was >24 U/mL.

### 2.5. Statistical Analysis

Data were reported as frequency (%), mean ± standard deviation (SD). Kolmogorov–Smirnov tests were performed to assess the distribution of normality for each data set. The independent *t*-test was used to analyze normally distributed variables to compare between cancer and control patients. Chi-square was used for group frequency comparison to find out the association. A *p*-value less than 0.05 (two-tailed) was considered statistically significant. SPSS software version 27 (IBM, Chicago, IL, USA) was used for all statistical analyses.

### 2.6. Ethical Approval

The ethical clearance was obtained from the authority of Enam Medical College and Hospital, from where the samples of the patients were collected (Ref: 26-11-2019).

## 3. Results

A total of 80 individuals were recruited in this study, including 40 cancer patients and 40 age- and sex-matched healthy controls. There was a predominance of male individuals (65%) in both the cancer and control groups. The most common cancer site was the liver, involving 37.5% (n = 15) of the patients. Among the cancer patients, 72.5% (n = 29) were in the early stage, with more than half of the patients (n = 21) undergoing chemotherapy. Diabetes (n = 13) and obesity (n = 12) were the most common comorbidities among the cancer patients (Table 1).

Anticardiolipin antibodies were tested positive in 60.0% (n = 24) of the cancer patients compared to 0% in the healthy controls (*p* < 0.001). No significant difference was observed between males and females in terms of the presence of aCL antibodies. The mean aCL concentration in cancer patients was 59.7 ± 49.2 U/mL, with the highest concentration of 133.0 U/mL in a 50-year-old male patient with early stage liver cancer. On the other hand, the concentration of aCL antibodies was 2.3 ± 2.0 U/mL in the healthy controls (*p* < 0.001). The aCL concentration ranged from 89.0 to 133.0 U/mL among the aCL-antibody-positive cancer patients (Table 2 and Table 3).

All the patients with lung cancer (n = 6) were diagnosed with positive aCL antibodies (Table 4). Interestingly, the highest number (n = 9) of patients with liver cancer tested positive for aCL antibodies, representing 60% of these patients. Among the patients with advanced-stage cancer (n = 11), 72.7% were aCL-positive. Patients with early stage cancer (n = 29) exhibited a lower prevalence (55.2%) of aCL positivity (n = 16). About 81.8% (n = 21), 66.7% (n = 9), and 50.0% (n = 2) of the cancer patients undergoing surgical treatment, chemotherapy, and drug treatment tested positive for aCL antibodies, respectively. All cancer patients suffering from heart diseases tested positive for aCL antibodies (n = 9). Other conditions associated with positive aCL in cancer patients included smoking habit (80.0%), obesity (66.7%), diabetes (61.5%), and hypertension (33.3%).

The association of certain risk factors with high aCL concentration was evaluated by chi-square test, which revealed the cancer site colon to be a borderline significant risk factor for the presence of aCL antibodies (χ^2^ (1, n = 24) = 3.810, *p* = 0.051) (Table 4). Among the treatment options, surgical approach was associated with aCL positivity (χ^2^ (1, n = 24) = 3.009, *p* = 0.083). Among the comorbidities, heart diseases (χ^2^ (1, n = 24) = 7.742, *p* = 0.005) exhibited a significant association with the presence of aCL antibodies.

## 4. Discussion

Noncommunicable diseases now account for most global deaths, and cancer is one of the primary causes of death and the greatest barrier to increasing life expectancy in every country in the 21st century [1]. The preponderance of imaging techniques, including X-rays, magnetic resonance imaging, computed tomography, endoscopies, and ultrasounds, can only detect cancer after visible tissue changes have occurred [4,5]. A cancer biomarker is a measurable biological molecule that indicates the prevalence of cancer in the body. Biomarkers for cancer can be detected in blood as well as in other tissues or body fluids such as saliva and urine [10]. In this study, we explored the existence of aCL antibodies, which could be considered as a potential risk factor and as early detection tools for thrombotic events in cancer patients.

In this case–control study, we detected significantly higher prevalence and serum levels of aCL in patients with cancer compared to healthy controls. Previous studies also detected the prevalence of different aPLs in cancer patients with a varied frequency ranging from 5.7 to 74.0% [5,9,23,25]. Two case–control studies conducted on Spanish and Italian cancer populations showed 8.2-fold and 5.6-fold increases in aCLs. In both of these studies, aCL was the most prevalent antibody, detected in 51.5 to 60.9% of the aPL-positive cancer patients [21,23]. On the contrary, cross-sectional studies represented a high prevalence of other types of aPLs in cancer patients. Anti-β2-GPI IgA was obtained in 46.9% of the thrombotic cancer patients in a Singaporean population [9], and LA was found in 61.0% of the critically ill cancer patients in a Brazilian cohort [26]. This Brazilian study also reported the highest frequency of aPLs (74.0%) in cancer patients requiring intensive care unit support. The frequency of aCL was predominantly higher (60.0%) in our study compared to the studies conducted in Singapore [9], Brazil [26], Israel [25], Spain [21], and Italy [20,22], whereas aCL antibody was detected in 1.0–21.8% of the cancer patients.

A recent systematic review illustrated that the patients with gastrointestinal, genitourinary, and lung cancers have 5.1, 7.3, and 5.2 times augmented risk of having aCL, respectively, compared to the healthy controls [27]. Another study consisting of patients with solid and hematologic cancers illustrated that the patients with hematologic cancers were 1.67 times more likely to present aCL positivity than the patients with solid cancers. The prevalence of aCL among individuals with pancreatic cancer and Non-Hodgkin’s lymphoma was the highest among the solid and hematological malignancies, respectively [25]. The presence of moderate to high levels of aCL was confirmed by another recent study by Majdan et al. [28], where the patients with uterine malignancies were about two times more likely to be diagnosed with positive aCL compared to non-cancerous gynecological disease patients.

In our study, 100% of the lung cancer patients were detected with positive aCL, and colon cancer patients possessed a borderline significant risk of exhibiting aCL (*p* = 0.051). An Italian study found that non-metastatic colon cancer patients are five times more likely to be diagnosed with positive aCL compared to the healthy controls [24]. Bazzan et al. [23] identified that patients with advanced-stage cancer were more likely to present aPLs compared to early stage cancer patients. In alignment with that, we observed that the frequency of aCL positivity was 17.6% higher in advanced-stage cancer individuals compared to early stage cancer patients in our study.

The risk of thrombosis is several times higher in cancer cases compared to healthy individuals. Several biochemical pathways, including the ability of cancer cells to release procoagulants, fibrinolytic proteins, and inflammatory cytokines with host-cell-adhering properties, and some clinical risk factors, such as cancer therapies (i.e., chemotherapy and immunotherapy), cancer staging (i.e., advanced stage), and cardiovascular diseases, are believed to be responsible for a higher frequency of thrombosis in cancer. However, there is still ambiguity about the underlying cause of cancer-associated thrombosis [6,22]. An elevated level of aPLs was also observed following cancer surgery, which is consistent with our study. We observed that 81.8% of the patients who underwent surgery were diagnosed positively with aCL antibodies [29]. In other studies, 52.4% of the thrombotic cancer patients receiving chemotherapy were aPL-positive. It was identified as a risk factor for venous thromboembolism (VTE) and pulmonary embolism and surgery [26,30]. In our study, 66.7% of the patients receiving chemotherapy tested positive for aCL. A study conducted with 33 Asian cancer patients with thrombosis showed that anti-β2-GPI was the most prevalent aPL and was detected in 45.4% of the overall study population [9]. In another study, aCL was diagnosed positive in 10 out of 13 thrombotic cancer patients (77.0%); however, aCL was present in only 6.0% of the non-thrombotic cancer patients [25]. Furthermore, in a Spanish study, Font et al. [21] reported that 8.0% of the VTE patients were diagnosed positive for any of the aPLs, and only 1.4% of the non-VTE cancer patients were diagnosed positive. On the other hand, a cohort study exhibited an almost similar proportion of aPL among cancer patients with thrombotic complications (24.2%) and without thrombotic complications (24.0%) [26]. Similarly, in another study, aPL antibodies were not significantly associated with thrombotic complications in cancer patients, and 72.2% of the thrombotic solid cancer patients were positive for IgM aCL compared to 57.1% in non-thrombotic solid cancer cases in that study [31]. Therefore, whether aPL antibodies, including aCL antibodies, should be considered as a risk factor for thrombotic events in cancer patients or not is still inconclusive.

Our study has several strengths. To the best of our knowledge, it is the first study to identify the prevalence of aCL antibodies in patients with cancer compared to healthy controls in the Bangladeshi population. Nevertheless, there are some notable limitations. Data were collected from a single study center with a statistically needed robust sample size; hence, the result can vary with a large data volume or a multi-center study center. Due to limited funds, we used the kit to detect only IgG and IgM aCL and not to evaluate other aPLs or even anticoagulants. Another limitation of this study is that we were not able to carry out repeat testing to confirm the persistence of aCL, nor were we able to assess double- or triple-positive antiphospholipid antibodies. Therefore, future studies should be carried out with a larger sample size, comprising thrombotic data of cancer patients, precisely identifying serum levels of IgG and IgM aPL antibodies separately and evaluating cancer patients with double- or triple-positive aPLs.

## 5. Conclusions

Our case–control study identified significantly high frequency and serum levels of aCL antibodies in patients with cancer compared to healthy controls. In addition, a significant association was observed between the presence of aCL antibodies and advanced cancer stage, heart diseases, and surgical treatment strategies. Future studies should address the involvement of aCL antibodies in the pathogenetic mechanisms that could account for the development of thrombotic complications in cancer patients.

## Figures and Tables

**Table 1 cancers-15-02087-t001:** Characteristics of the patients.

Characteristics	Cancer Patients (%)	Healthy Controls (%)
Age (Mean ± SD)	54.9 ± 14.4	54.9 ± 6.1
Gender		
Male	26 (65)	26 (65)
Female	14 (35)	14 (35)
Cancer involving organs		
Liver	15 (37.5)	
Lung	6 (15.0)	
Prostate	5 (12.5)	
Colon	5 (12.5)	
Breast	2 (5.0)	
Oesophagus	2 (5.0)	
Ovary	2 (5.0)	
Retina	1 (2.5)	
Blood	1 (2.5)	
Urinary Tract	1 (2.5)	
Cancer stages		
Early stage (I and II)	29 (72.5)	
Advanced	11 (27.5)	
Treatments		
Chemotherapy	21 (52.5)	
Surgery	11 (27.5)	
Drug	4 (10.0)	
Comorbidities		
Diabetes Mellitus	13 (32.5)	
Obesity	12 (30.0)	
Hypertension	9 (22.5)	
Smoking	5 (12.5)	
Cardiac Problem	9 (22.5)	

**Table 2 cancers-15-02087-t002:** Profile of anticardiolipin antibodies in cancer patients and healthy controls.

aCL Antibodies	Cancer Patients (%)	Healthy Controls (%)	t (df)	*p*-Value
Present	24 (60.0)	0 (0.0)	−7.649 (78)	**<0.001**
Absent	16 (40.0)	40 (100.0)		
Concentration (U/mL) Mean ± SD	59.7 ± 49.2	2.3 ± 2.0	7.377 (78)	**<0.001**

t (df): *t*-test (degree of freedom); aCL: anticardiolipin.

**Table 3 cancers-15-02087-t003:** Individual profile of anticardiolipin-antibody-positive cancer patients.

Gender	Age (Years)	Cancer Site (Stage)	Anticardiolipin Antibody Concentration (U/mL)
Male	58.0	Lung (E)	89.0
Male	55.0	Liver (E)	100.0
Male	50.0	Liver (E)	113.0
Male	60.0	Liver (E)	90.0
Female	45.0	Lung (E)	92.0
Female	52.0	Liver (E)	97.0
Male	45.0	Esophagus (E)	90.0
Female	43.0	Esophagus (A)	90.0
Female	46.0	Breast (E)	93.0
Male	58.0	Lung (A)	94.0
Male	60.0	Prostate (E)	94.0
Male	60.0	Liver (E)	110.0
Female	50.0	Liver (E)	104.0
Male	40.0	Colon (E)	89.0
Female	65.0	Liver (A)	98.0
Male	22.0	Urinary tract (E)	94.0
Male	40.0	Lung (A)	115.0
Male	50.0	Liver (E)	133.0
Male	80.0	Prostate (E)	93.0
Male	70.0	Lung (A)	101.0
Female	50.0	Liver (A)	93.0
Female	65.0	Ovary (A)	91.9
Female	67.0	Lung (A)	115.0
Male	18.0	Retina (E)	92.0

E = early stage (I and II), A = advanced stage (III and IV).

**Table 4 cancers-15-02087-t004:** Association of the presence of anticardiolipin antibodies and risk factors in patients with cancer (n = 24).

Characteristics	Frequency (%)	χ^2^ (df)	*p*-Value
Age (Mean ± SD)	52.1 ± 14.0	22.292 (22)	0.443
**Cancer involving organs**			
Liver	9 (37.5)	0.165 (1)	0.685
Lung	6 (25.0)	0.938 (1)	0.333
Prostate	2 (8.3)	0.952 (1)	0.329
Colon	1 (4.2)	3.810 (1)	0.051
Breast	1 (4.2)	0.088 (1)	0.767
Esophagus	2 (8.3)	1.404 (1)	0.236
Ovary	1 (4.2)	0.088 (1)	0.767
Retina	1 (4.2)	0.684 (1)	0.408
Urinary tract	1 (4.2)	0.684 (1)	0.408
**Cancer stages**			
Early	16 (66.7)	1.024 (1)	0.312
Advanced	8 (33.3)	1.024 (1)	0.312
**Treatments**			
Chemotherapy	14 (58.2)	0.819 (1)	0.366
Surgery	9 (37.5)	3.009 (1)	0.083
Drug	2 (8.3)	0.185 (1)	0.667
**Comorbidities**			
Diabetes	8 (33.3)	0.019 (1)	0.890
Obesity	8 (33.3)	0.317 (1)	0.573
Hypertension	3 (12.5)	3.441 (1)	0.064
Smoking	4 (16.7)	0.952 (1)	0.329
Heart diseases	9 (37.5)	7.742 (1)	0.005

**χ**^2^ (df): Chi-square (degree of freedom).

## Data Availability

All data are available upon request to the corresponding authors.

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
