# Peer review of "Anticardiolipin Antibodies in Patients with Cancer: A Case–Control Study"

_cancers, 2023, doi:10.3390/cancers15072087_

Round 1
Reviewer 1 Report
In this MS by Md. Ashraful Islam Nipu et al., the authors aimed to explore the presence of aCL antibodies in patients with different types of 80 cancer compared to that in healthy control subjects
The topic is interesting, in order to improve the MS I have the following major comments and suggestions:
A summarized scheme with all the steps included in this study is very useful for readers for a better understanding of MS.
I suggest keeping a structured abstract but without writing the details: (1)Background, (2)Methods, (3)Results.
Line 43: correct “antibody” with “antibodies”
Line 48: add a reference
Lines 54-55: the authors stated: “The treatment strategies of cancer, including surgery, cytotoxic chemotherapy, and hormone therapy, also increase the risk of thrombosis and early mortality” - Really? Do all these therapeutic strategies used in the pharmacotherapeutic management of cancer in oncological patients lead to their early death? Mention evidence-based mechanisms of action for the statements made.
Line 56: “The thrombotic events in cancers are not fully understood” – it is not exactly true, add more details and appropriate references.
Lines 58-60: The authors contradict themselves with what they mentioned previously. Please revise.
Line 88: add the number and date and place for ethical consent regarding the enrolled patients in this study.
Discussion section: the comparisons with the specialized literature are not extensive enough. Mention more limitations of this study.
What perspectives for human health does this MS have?
Consider revision accordingly.
Author Response
Author response: Reviewer 1
In this MS by Md. Ashraful Islam Nipu et al., the authors aimed to explore the presence of aCL antibodies in patients with different types of 80 cancer compared to that in healthy control subjects. The topic is interesting, in order to improve the MS I have the following major comments and suggestions:
Points 1: A summarized scheme with all the steps included in this study is very useful for readers for a better understanding of MS.
Author response 1: Many thanks for your suggestion. We’ve added a graphical abstract for better understanding. We hope it’s alright now.
Points 2: I suggest keeping a structured abstract but without writing the details: (1)Background, (2)Methods, (3)Results.
Author response 2: Many thanks for your suggestion. We have corrected this issue.
Points 3: Line 43: correct “antibody” with “antibodies”
Author response 3: Thank you, corrected.
Points 4: Line 48: add a reference.
Author response 4: Added and the definition has also been improved. Thank you.
Points 5: Lines 54-55: the authors stated: “The treatment strategies of cancer, including surgery, cytotoxic chemotherapy, and hormone therapy, also increase the risk of thrombosis and early mortality” - Really? Do all these therapeutic strategies used in the pharmacotherapeutic management of cancer in oncological patients lead to their early death? Mention evidence-based mechanisms of action for the statements made.
Author response 5: We apologise and would like to state that the sentence was not written in the appropriate tone and that is why it sounded misleading. What we meant was that, generally cancer patients are more prone to thrombotic complications. Some treatment procedures also stimulate thrombosis as an adverse effect. Thrombosis leads to early death of cancer patients. It is undoubtedly not true that the drugs promote early death. We have toned down and clarified the statement. Thank you.
Points 6: Line 56: “The thrombotic events in cancers are not fully understood” – it is not exactly true, add more details and appropriate references.
Author response 6: Many thanks for your comment. We have revised the sentence carefully.
Points 7: Lines 58-60: The authors contradict themselves with what they mentioned previously. Please revise.
Author response 7: Agreed, as we firstly said that the events are not fully understood, but on the very next line, we briefly added the mechanism. Now, we’ve added some details about thrombotic events and cancer and that is why those lines hopefully would not sound contradictory now. Thank you.
Points 8: Line 88: add the number and date and place for ethical consent regarding the enrolled patients in this study.
Author response 8: We’ve added details of ethical consent in method section 2.6. Thank you.
Points 9: Discussion section: the comparisons with the specialized literature are not extensive enough. Mention more limitations of this study. What perspectives for human health does this MS have? Consider revision accordingly.
Author response 9: Many thanks for your suggestions. In the revised mansucript, we have added some more appropriate discussion sections and acknowledged some additional limitations of this study. Thank you.

Reviewer 2 Report
1. The English can be improved; I have provided some suggestions below, but will be best that the manuscript be edited by a native English writer. (a) In general, avoid use of ‘it’, as ‘it’ is ambiguous; eg line 50: use ‘cancer’ instead of ‘it’; (b) line 69: “…antiprothrombin antibodies which also play a role…”delete word ‘which’. (c) line 186: “times of augmented” delete word ‘of’. (d) line 196: “patients have detected with positive aCL” replace word ‘have’ with ‘were’ (e) line 214: What is ‘it’?
2. Line 60: “…the cancer cells enter a hypercoagulable state”. It is the patient who enters a hypercoagulable state, as facilitated by the cancer.
3. Line 65: aPL “…antibodies that are erroneously produced as an autoimmune response to phospholipids.” The term aPL is something of a misnomer in that the antibodies are predominantly directed against phospholipid (PL)-binding proteins in complex with negatively-charged PLs rather than PL per se.
4. Lines 110-112: “Then 100 μl conjugate solution was added into each well, and the plate was again incubated for 30 minutes at room temperature (250C). 100 μl substrate solution…” Isn’t there another wash step after incubation with the conjugate solution?
5. Tables 1 & 2: “Age *” why have an ‘*’ here that pints to a footer; use “Age [Mean ± SD]” instead
6. Lines 231-239. Limitations. The study has several additional limitations: The authors did not evaluate other aPL or even lupus anticoagulant; they did not perform repeat testing to confirm persistence of the aPL; they did not assess for double positive or triple positivity for aPL
Author Response
Author response: Reviewer 2
Points 1: The English can be improved; I have provided some suggestions below but will be best that the manuscript be edited by a native English writer. (a) In general, avoid use of ‘it’, as ‘it’ is ambiguous; eg line 50: use ‘cancer’ instead of ‘it’; (b) line 69: “…antiprothrombin antibodies which also play a role…”delete word ‘which’. (c) line 186: “times of augmented” delete word ‘of’. (d) line 196: “patients have detected with positive aCL” replace word ‘have’ with ‘were’ (e) line 214: What is ‘it’?
Author response 1: Many thanks for your suggestions. All of the suggestions are appreciated and have already been addressed and corrected in the manuscript using track changes. Thank you.
Points 2: Line 60: “…the cancer cells enter a hypercoagulable state”. It is the patient who enters a hypercoagulable state, as facilitated by the cancer.
Author response 2: Many thanks for your suggestions. All of the suggestions are appreciated and have already been addressed and corrected in the manuscript using track changes.
Points 3: Line 65: aPL “…antibodies that are erroneously produced as an autoimmune response to phospholipids.” The term aPL is something of a misnomer in that the antibodies are predominantly directed against phospholipid (PL)-binding proteins in complex with negatively charged PLs rather than PL per se.
Author response 3: Many thanks for your suggestions. All of the suggestions are appreciated and have already been addressed and corrected in the manuscript using track changes.
Points 4: Lines 110-112: “Then 100 μl conjugate solution was added into each well, and the plate was again incubated for 30 minutes at room temperature (250C). 100 μl substrate solution…” Isn’t there another wash step after incubation with the conjugate solution?
Author response 4: Thank you. Your suggestion has been appreciated and corrected by adding the wash step information.
Points 5: Tables 1 & 2: “Age *” why have an ‘*’ here that pints to a footer; use “Age [Mean ± SD]” instead
Author response 5: Thank you for indicating the issue. This has been corrected.
Points 6: Lines 231-239. Limitations. The study has several additional limitations: The authors did not evaluate other aPL or even lupus anticoagulant; they did not perform repeat testing to confirm persistence of the aPL; they did not assess for double positive or triple positivity for aPL
Author response 6: Many thanks for your suggestions. We have added a number of limitations based on suggestions of reviewer 1 and reviewer 2. We hope this section reads better now. Thank you.

Round 2
Reviewer 1 Report
No answer given.
Reviewer 2 Report
I still think the English language needs improving; I have no further suggestions for the scientific content